# A Review of Properties of Nanocellulose, Its Synthesis, and Potential in Biomedical Applications

**Aayushi Randhawa** [1,2,†], **Sayan Deb Dutta** [1,†], **Keya Ganguly** [1], **Tejal V. Patil** [1], **Dinesh K. Patel** [3] **and Ki-Taek Lim** [1,2,3,*]

[1] Department of Biosystems Engineering, Kangwon National University, Chuncheon 24341, Korea; randhawaaayushi15@gmail.com (A.R.); sayan91dutta@gmail.com (S.D.D.); gkeya14@gmail.com (K.G.); tejal.patil07@gmail.com (T.V.P.)

[2] Interdisciplinary Program in Smart Agriculture, Kangwon National University, Chuncheon 24341, Korea

[3] Institute of Forest Science, Kangwon National University, Chuncheon 24341, Korea; dbhu10@gmail.com

\* Correspondence: ktlim@kangwon.ac.kr

† These authors contributed equally to this work.

**Abstract:** Cellulose is the most venerable and essential natural polymer on the planet and is drawing greater attention in the form of nanocellulose, considered an innovative and influential material in the biomedical field. Because of its exceptional physicochemical characteristics, biodegradability, biocompatibility, and high mechanical strength, nanocellulose attracts considerable scientific attention. Plants, algae, and microorganisms are some of the familiar sources of nanocellulose and are usually grouped as cellulose nanocrystal (CNC), cellulose nanofibril (CNF), and bacterial nanocellulose (BNC). The current review briefly highlights nanocellulose classification and its attractive properties. Further functionalization or chemical modifications enhance the effectiveness and biodegradability of nanocellulose. Nanocellulose-based composites, printing methods, and their potential applications in the biomedical field have also been introduced herein. Finally, the study is summarized with future prospects and challenges associated with the nanocellulose-based materials to promote studies resolving the current issues related to nanocellulose for tissue engineering applications.

**Keywords:** nanocellulose; additive manufacturing; nanocomposites; biomedical applications





## 1. Introduction

Congenital abnormalities, trauma, malfunctioning, aging, disease, and functional loss of organs or tissues are the most common public health challenges [1,2]. Biopolymers are obtained from natural sources and are biocompatible with the human body. Chitosan, alginate, starch collagen, cellulose, and other biopolymers have been found in many natural species [3,4].

Cellulose is an abundant, sustainable, naturally occurring biopolymer that has aroused attention in the biomaterial community to produce eco-friendly and biocompatible goods [5,6]. These naturally occurring polymers have attracted significant interest in biomedical applications such as the administration of drugs, wound healing, and scaffold tissue engineering [7]. Cellulose is a naturally occurring linear molecule made up of D-anhydroglucopyranose subunits linked via β-glycosidic bonds, in which cellobiose is the repetitive unit [8,9]. The term nanocellulose (NC) refers to cellulose with a dimension in the nanoscale [10]. Obtained from native cellulose, nanocellulose occurs mainly in plants, animals, and bacteria in three primary forms: (1) cellulose nanocrystals (CNC), (2) cellulose nanofibers (CNF) which are sometimes also known as nanofibrillated cellulose (NFC), and micro-fibrillated cellulose (MFC), and (3) microbial or bacterial nanocellulose (BNC) [11,12]. The molecular backbone of cellulose is similar to all three types of nanocellulose; however, chemical and physical characteristics may differ based on the origin and purification procedures [13].

The sources of bacterial cellulose (BC) or bacterial nanocellulose (BNC) include mainly the Gram-negative bacteria *Acetobacter xylinum* (*Glucanacetobacter xylinus*). *Archromobacter*, *Acetobacter*, *Alcaligenes*, *Sarcina*, *Pseudomonas*, and *Rhizobium* are the microbial sources capable of producing nanocellulose [14–19]. Maize cob, cotton, wheat bran, banana leaves, sugar beet, wood, potato tuber, and mulberry husks are potential plant sources for nanocellulose extraction [20,21]. CNC is primarily obtained by the acid hydrolysis of natural cellulose [22–24]. The CNC is 100–500 nm long and 4–20 nm wide, with crystalline syringe-like structures. CNF is frequently produced from biomass degradation and chemical reactions. CNF is a 1 μm long combination of amorphous and crystalline, having a diameter of 20–100 nm, and is the purest type of nanocellulose without any contaminating components [19,25–27].

Nanocellulose offers tunable surface functionalization, excellent mechanical strength, high hydrophilicity, and biocompatibility. It can also easily create hydrogen bond networks [13,28–34]. Therefore, nanocellulose is receiving a deep interest in biological applications, where it can be used as a carrier for the successful delivery of drugs to damaged tissues and other biomedical applications [35–37]. In the current study, we aim to provide a detailed study on the characteristics of nanocellulose, their production, and fabrication methods of nanocellulose-based composites by using additive manufacturing techniques. Moreover, the biomedical applications of nanocellulose-based composites have been discussed. In the end, future trends, challenges, and possible opportunities for nanocellulose in the biomedical field have also been highlighted. Figure 1 represents a schematic visual of the current review paper.

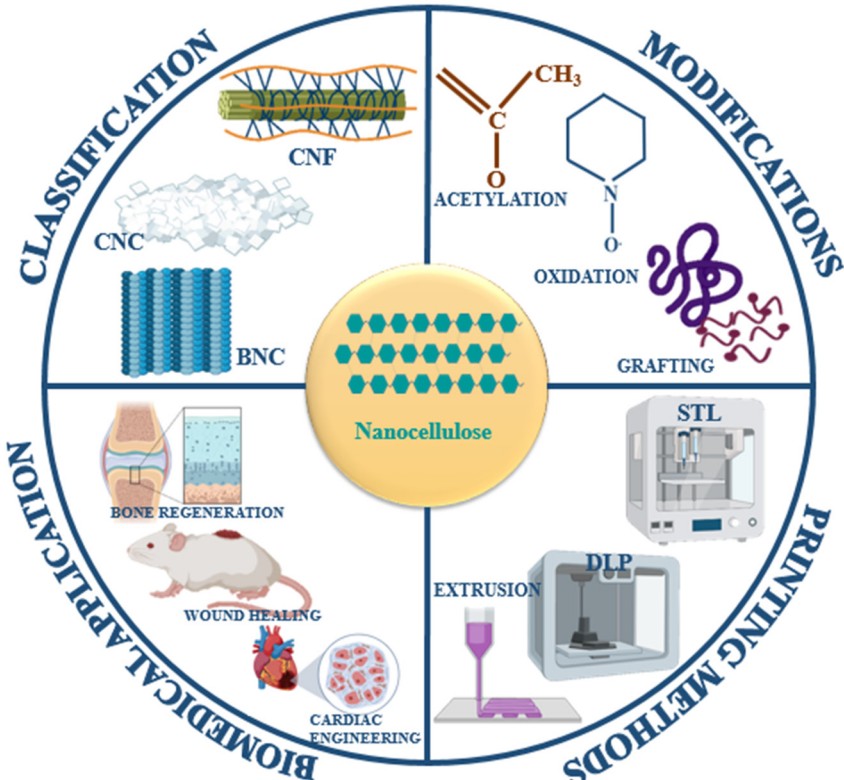

**Figure 1.** Schematic representation of the classification, modifications, printing methods, and biomedical applications of nanocellulose and its derivatives.

## 2. Classification and Synthesis of Nanocellulose

### 2.1. Cellulose Nanocrystal (CNC)

Acid hydrolysis using mineral acids such as phosphoric, sulfuric, or hydrochloric acid and methods such as enzymatic procedures are commonly known for isolating CNC [38–42]. The amorphous region of cellulose fibers is cleaved during hydrolysis, resulting in a rigid,

highly crystalline, and rod-like nanostructure. During the acid hydrolysis process, the incorporation of acid leads the charge on the surface of CNC, allowing colloidal dispersion to form quickly [40]. Based on the source of cellulose, the crystalline percentage can vary from 50–90%, and the elastic moduli have been determined to be between ~105−168 GPa [43,44]. The longest type of CNC can be obtained from tunicates, having a length of 100–300 nm and 5–50 nm in diameter [40]. CNC shows a shear-thinning property and functions like liquid crystals, creating asymmetrical nematic phases at specific concentrations [45]. When CNC is introduced into polymer materials, hydrogen bond networks emerge, allowing for stress transfer, which is highly intriguing in additive manufacturing [46].

### 2.2. Cellulose Nanofiber (CNF)

By-products of rice, corn, barley, banana, wheat, and sugar cane are used for CNF synthesis. Furthermore, wood and wood pulp are the most common sources of nanocellulose employed for manufacturing CNF [47].

Mechanical treatment is the most common method for isolating CNF. However, other methods, such as enzymatic and chemical treatments, have also been documented [12,21]. High-pressure homogenization, ultrasonic fiber delamination, and ball-mining are the preferred approaches to obtaining CNF. A dilute fiber–water mixture is supplied via a narrow, high-pressure outlet during the homogenization process. A significant change in pressure promotes fibrillation. Microfluidization is identical to homogenization, except there are no moving elements in the microfluidizer, resulting in less blockage. The shear force decomposes the fibers in the interaction chamber, operating on the channel wall and collision current. To manufacture CNF, the fiber–water solution must be homogenized multiple times, which costs a lot of energy [12].

In contrast to the homogenizer, the micro-fluidizer runs at a constant speed, while the homogenizer maintains a constant process mass. The threads are connected through a gap between the spinning and stationary discs during refining, performed using a disc filter [48]. The oxidation of cellulose using 2,2,6,6-tetramethylpiperidine-1-oxyl radical (TEMPO) is a well-known chemical pretreatment procedure for the production of CNF. Strong nitroxyl radical, TEMPO, has been found to oxidize alcohols to carboxylate groups on the surface of carbohydrates via aldehydes and is considered for its reaction rate and regioselectivity [49].

Compared to CNC and BNC, CNF is a flexible and short fibril, with a length of 1–10 μm and a diameter range within 20–100 nm, which varies depending on the defibrillation process and the cellulose sources. The crystallinity of CNF is substantially lower than that of BNC and CNC; however, CNF is made up of crystalline and amorphous domains. The Young's modulus of CNF is approximately 30 GPa, which is less than the BNC and CNC. CNF accounts for thixotropic and shear-thinning properties, although it is often unstable in solution [50].

### 2.3. Bacterial Nanocellulose (BNC)

Bacterial nanocellulose (BNC) is manufactured through the bottom-up technique (glucose molecules are metabolized to produce nanocellulose), whereas the other types of nanocellulose, such as CNC and CNF, are prepared through a top-down approach (cellulose molecules of higher dimensions are broken down to produce nanocellulose) [51]. BNC is produced as floccus, which then emerges into fiber and interweaves to create pellicles. Species of bacteria such as *Acetobacter*, *Azotobacter*, *Pseudomonas*, and *Sarcina ventriculi* are capable of producing BNC. The fermentation of glycerol and glucose contained in natural or synthetic media (in which the bacteria is cultivated) results in BNC production. The mechanical characteristics, crystallinity, and shape of the produced BNC are influenced by cultivation conditions such as oxygen delivery, pH variations, etc. [35].

BNC is biodegradable and commonly manufactured in the form of 10–100 nm diameter fibers, with a crystallinity range within 75–96% [52,53]. Several approaches have been used

to calculate the mechanical characteristics of the single fiber, with Young's moduli ranging from 78 to 114 GPa, which is higher than CNF and CNC [54,55].

Figure 2 depicts the desirable properties such as lightweight, high flexible nature, and non-toxicity of various forms of nanocellulose derived from the cellulose, which makes them potential candidates in the biomedical field.

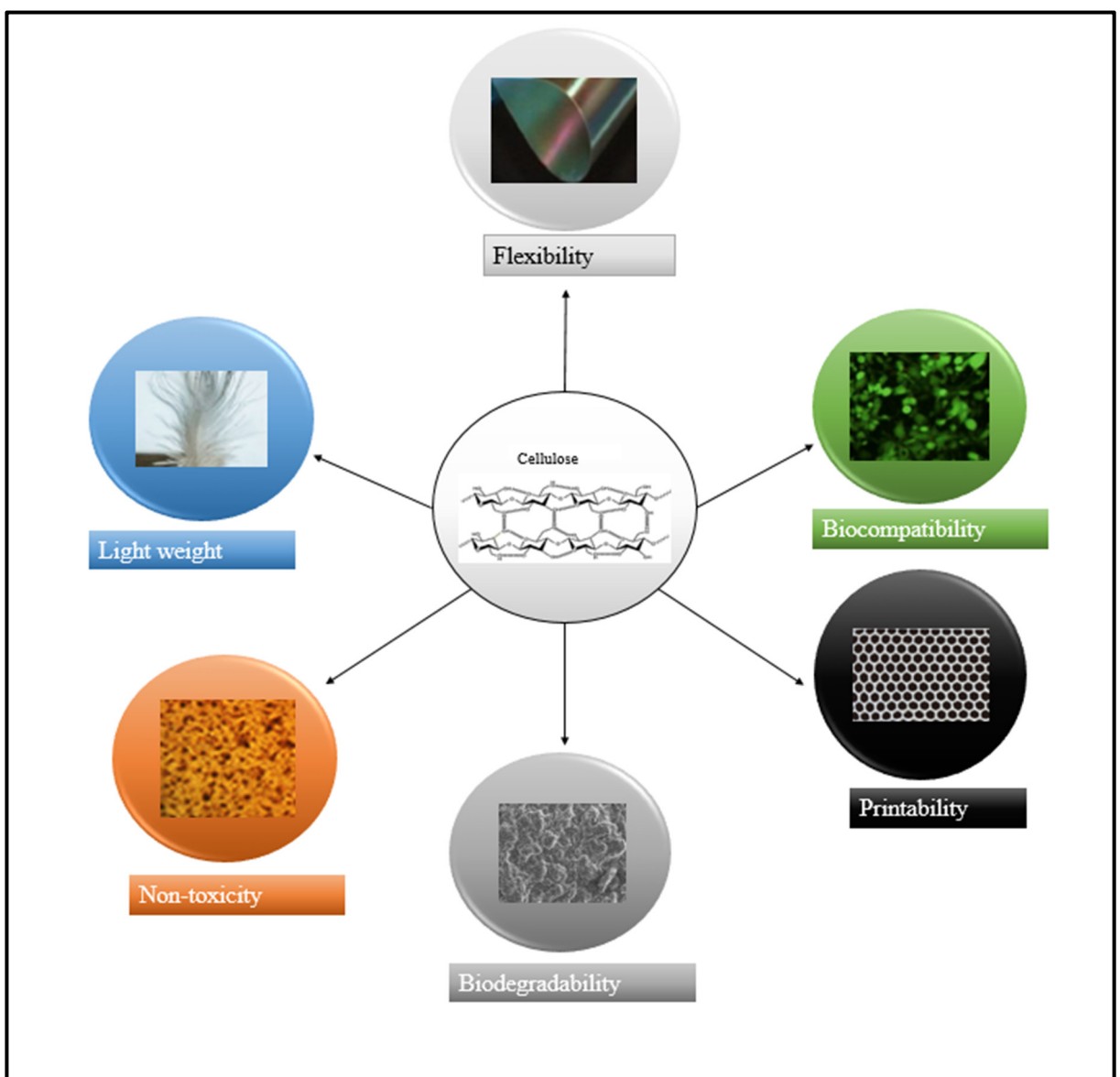

**Figure 2.** The attractive characteristics of nanocellulose, including its biocompatible nature, high precision printing, biodegradability, low toxicity, lightweight, and flexibility, are crucial for synthesizing composites and hydrogel fabrication. Adapted with permission from Refs. [56–62].

## 3. Chemical Modification and Functionalization of Nanocellulose

For a particular application, the surface functionalization of cellulosic nanostructure offers a possible great platform. The surface properties of cellulose-based materials can be successfully modified. Various chemical alteration methodologies are performed to improve the effectiveness of the isolation method and to modify the hydrophobicity of nanocellulose materials. These alterations in the nanocellulose materials enhance the degradability and compatibility of nanocellulose [63].

### 3.1. Ionic Charge Transfer to Nanocellulosic Surfaces

The 2,2,6,6-Tetramethylpiperidine-1-oxyl (TEMPO)-assisted oxidation is often employed to make the surface of nanocellulose hydrophobic, as well as a pretreatment for the isolation of nanofibers. This method was first reported by De Nooy et al. [64], demonstrating that only the primary hydroxymethyl group of polysaccharides are oxidized by TEMPO, while the secondary hydroxyl group remains intact. This approach transforms the glucose unit's C6 alcohol functionalities into carboxylic acid [65,66]. When $NaOCl_2$/NaOCl/TEMPO is used to catalyze the oxidation of cellulose, the oxidation at the 6th position forms anionic carboxylate, which produces a high charge level, resulting in enhanced water dispersibility [67]. The negatively charged ions formed through this oxidation method on the surface of CNC significantly raise the electrostatic repulsion. Upon hydrolyzing the cellulose fibers with HCl, Araki et al. demonstrated TEMPO-catalyzed oxidation of CNC [38]. The morphological characteristics of the oxidized CNC were found to be the same as that of the CNC substrate and were quickly dissolved in the water.

### 3.2. Production of Hydrophobic Surface on the Nanocellulosic Materials

The plasticization of lignocellulosic fibers occurs when cellulosic alcohols are acetylated [68]. To improve the hydrophobicity of cellulosic fibers, acetylation is often used. To catalyze the acetylation of nanocellulose, dry acetic acid and acid anhydride are gradually added, followed by sulfuric acid. Sassi and Chanzy first proposed the two primary acetylation processes [69]. The two processes, fibrous and homogenous, depend on the swelling diluent's availability or lack. In the fibrous method, a diluent such as toluene is added to the reaction solution to make the acetylated cellulose insoluble. The native structure is retained while a higher level of acetylation is attained. Acetylated chains are soluble in sulfuric acid and acetic acid-containing medium in a diluent-free homogenous method. As a result, after considerable acetylation, the cellulose substrate shows significant structural alterations.

For the acetylation of mechanically separated NFC, Bulota and colleagues utilized acetic anhydride [70]. In their study, they demonstrated that nanofibers with a higher level of substitution impacted the characteristics of the polylactic acid-acetylated NFC composite significantly. The estimated contact angle was increased from 33° (for non-acetylated) to 115° (for acetylated) nanofibers, indicating a significant improvement in hydrophobicity. As an acyl donor, the enzyme lipase from *Aspergillus niger* was recently used to acetylate NFC via acetic anhydride [71]. Compared to the acetylation via the chemical method, the enzyme acetylation on NFC substantially increased hydrophobicity [71].

### 3.3. Cellulose Graft Copolymerization

This is an interesting and adaptable way of adding several functional groups to the polymer to change the physical and chemical characteristics [72]. The graft copolymerization method enables the effective qualities of two polymeric units to be combined in a single physical unit [73].

The graft copolymerization of cellulose is commonly accomplished by grafting a polymeric branch to the cellulose material, which induces certain qualities without compromising the natural features of the substrate. The method of grafting monomers to the cellulose and its derivatives usually occurs by a range of techniques that are divided into (a) free radical polymerization, (b) ionic and ring-opening polymerization, and (c) live radical polymerization [74].

"Grafting-to", "grafting-from", and "grafting-through" are the three main approaches on which the method of graft copolymerization relies [75]. In the "grafting-to" method, peptides or polymers are linked to the cellulose backbone, which involves fusing the polymer's reactive terminal end group to the hydroxyl group of the cellulose. Polystyrene, poly(caprolactone), and polypropylene are polymers that can be produced and connected to cellulose. The "grafting-from" approach involves first functionalizing the cellulose with an initiator and afterward polymerizing monomers straight from the surface. Compared

to the "grafting-to approach", this approach obtains increased polymer densities. In the "grafting-through" method, the functionalization of cellulose or its derivatives occurs with polymerizable vinyl-containing monomers. After that, the functionalized cellulose is combined with a co-monomer, followed by the polymerization initiation [74,76,77].

Dimensional stability, wrinkle recovery, elasticity ion exchange capabilities, thermal responsiveness, and microbiological invasion resistance can all be achieved depending on the polymer used for the grafting [78–82].

## 4. Nanocellulose-Based Composites

Nanocellulose possesses strong reinforcing qualities and is frequently utilized to create composites. Composites are made up of two or more phases (matrix and reinforcing) that have been combined to form a single new entity. There is a defined interface between these phases, which differ in their physicochemical characteristics. If a biopolymer is employed in manufacturing, the composite is termed a biocomposite. In nanocellulose-based composites, polymer serves as the matrix phase and nanocellulose as the reinforcing phase. In contrast to the matrix phase, the reinforcement phase makes up a more significant proportion of the composite [83].

Various composites, including sheets, paper, and film, have been produced differently. Interestingly, the inclusion of a modest percentage of nanocellulose contributes to a notable improvement in the properties of the composite due to the considerable surface area of the nanocellulose filler [84].

### 4.1. Polyvinyl Alcohol/Nanocellulose

The combination of nanocellulose and polyvinyl alcohol (PVA) is crucial for developing green bio-nanomaterials. In order to create CNC-PVA film by solution casting, Jasmine et al. isolated nanocrystalline cellulose (NCC) from *Acacia mangium* and incorporated it into the PVA film. Upon adding 2%, 5%, 7%, and 10% of NCC, the film's tensile strength was enhanced by 30%, 35%, 38%, and 50% compared with pure PVA film [85]. By employing different PVA/CNC composites membrane ratios, Jahan and co-workers investigated the composite's thermodynamic, mechanical, and swelling properties. They reported that the composite membrane's elastic modulus and tensile strength were proportional to the amount of CNC at an increased relative humidity. However, incorporating CNC can marginally lessen the thermal stability of the membrane [86]. Rescignano et al. developed CNC/PVA nanocomposite by combining poly (D,L-lactide-co-glycolide) nanoparticles. The inclusion of CNC improved the elongation characteristics and Young's modulus of the membrane, these properties making this nanocomposite a potential tool for drug delivery [87]. Enayati et al. addressed the function of nano-hydroxyapatite and CNF as fillers in electrospun PVA nanofibers-based composite scaffold for bone tissue regeneration. The presence of fillers increased the cellular activity of the scaffold and affected the in vitro degradation in phosphate-buffered saline. The outcomes supported the fibrous scaffold for tissue engineering [88].

### 4.2. Chitosan/Nanocellulose

Bacterial cellulose (BC) is a versatile material that holds promise for use in various biomedical and cosmetic applications because of its mechanical strength, shape, non-toxicity, biocompatibility, and chemical controllability. The BNC-based composites with other constituents such as synthetic and natural polymers allow the production of several biomedical goods [89]. In situ and ex situ approaches are widely employed for BNC composite synthesis [90]. The in situ method incorporates reinforcing materials into the culture media to aid BC synthesis; these materials eventually become a part of the generated hydrogel. For the ex vivo approach, composites are formed by impregnating or incorporating reinforcing components into synthetic polymer [91]. Both in situ and ex situ techniques have been used to prepare a composite of chitosan (Ch) and BC; the presence of O-H and N-H groups and structural similarities induce strong bonding between Ch and BC, and the

composite subsequently exhibits notable enhancement in biological and physicomechanical properties [92–94]. The remarkable biological qualities of gelatin have been studied for its use with scaffold material such as BC. BC–gelatin composites are produced via in situ and ex situ approaches. The composites showed improved cell proliferation and adhesion properties and have been employed for biomedical applications. Several polymers, including collagen, alginate, and Novo aloe vera, can be used with BC to broaden its therapeutic potential [91,95,96]. These composites with polymeric materials are widely used to confer antifungal, antimicrobial, and tissue regeneration properties. By impregnating BC sheets into an $AgNO_3$ solution, Maneerung et al. developed a BC–Ag composite. Nanoparticles adhered to the BC surface were examined with XRD spectral peaks and UV absorption. The BC–Ag composite exhibited extremely potent antibacterial activities against Gram-positive and Gram-negative bacteria [97]. Hence, the study indicated that incorporating metals and metal oxides into the BC composites imparts electrical conductivity and antimicrobial properties [97–100].

*4.3. Graphene/Nanocellulose*

Graphene is a thin, thermally, and electrically conductive transparent sheet. It is the fundamental building component of allotropes of carbon, including fullerenes, graphite, and carbon nanotubes [101–103]. Without the chemical functionalization of graphene, incorporating cellulose nanoparticles improves the graphene nanoparticle's dispersion in the aqueous environment and restricts aggregation [104]. The fabrication of nanocellulose/graphene composites begins with water-based dispersion. Different approaches such as filtration, filtration with hot pressing, freeze-casting, and freeze-drying can be employed for composite fabrication. Furthermore, other fabricating methods include incorporating graphene during the bacterial synthesis of nanocellulose and depositing graphene on a nanocellulose layer [105–112]. Several types of nanocellulose and graphene can be used for generating nanocellulose/graphene composite, and these composites can be supplemented with different compounds, for example, ceramic nanoparticles, carbides, oxides, enzymes, and polymers, to modify their characteristics for the particular application. An enhanced osteogenic differentiation is observed in human umbilical cord mesenchymal stem cells after adding graphene oxide to the electrospun cellulose acetate nanofibrous scaffold. It also improved these cells' growth and adhesion properties [113]. Nanocellulose/graphene composites offer considerable promise for developing wound dressing and antibacterial textiles due to their mechanical and antibacterial properties. Antibacterial textile fabricated via electrospinning a solution of graphene oxide sheets, cellulose acetate, and $TiO_2$ results in increased antibacterial activity against *Bacillus cereus* and *Bacillus subtilis* [114].

## 5. Additive Manufacturing in Printing Nanocellulose Composites

Tissue engineering was first proposed in the early 1990s when improved materials were developed to promote the production of neotissues, which are competent to replace or repair damaged tissues/organs [115]. Tissue engineering started to employ additive manufacturing technology in the previous decade. Additive manufacturing offers a quick and highly reliable way to make tissue-specific structures by regulating the intrinsic properties of the designed material, such as the shape, pore size, and deposition of multiple cell types and growth factors to imitate the original characteristics of the target tissue or organ [116]. Manufacturing processes that produce a solid three-dimensional structure from the computed data in a layer-by-layer fashion are referred to as the additive manufacturing technique [117,118]. The printing mechanisms of various additive manufacturing processes such as stereolithography, digital light processing, extrusion-based printing, and electrowriting are depicted in Figure 3.

*5.1. Stereolithography and Digital Light Processing (SLA and DLP)*

SLA and DLP printing techniques rely upon a top-down or a bottom-up approach to print light-sensitive polymeric resin, which is precisely polymerized and cured using a laser

or light irradiation. As per the two-dimensional patterned layer in a 3D CAD model, the printing platform can be immersed in the liquid resin, and the first layer can solidify on its surface after the light-induced curing by using a laser or a digital light projector [119]. The type of photoinitiator regulates the shape fidelity, the extent of curing, and the polymers used. In SLA printing, a build platform is placed closed to the surface of the resin, enabling a thin film of resin to form on it. A laser then generates the first layer of the object. After printing the first layer, the build platform slowly moves down, allowing the generated layer to be immersed in the resin and printing a new layer above it. The printed layers adhered to one other due to the crosslinking reaction until the final printed structure is obtained [120,121].

The DLP printing relies on a digital light projector to cure the liquid resin. At the bottom of the resin, a projector flashes visuals of the layers to a digital micromirror device, which precisely resends the light to the resin. As a result of redirecting an image projection by digital micromirror devices, the printing time is shorter than the SLA method [122,123]. Sun and colleagues demonstrated the use of the SLA technique to produce a CNF construct for optics and bio-adhesion [124]. The inverted SLA technique was employed to fabricate a poly(N-isopropylacrylamide)/CNF hydrogel with fine optics and bio-adhesion characteristics. With the addition of 2.0% mass CNF, which has controlled bio-adhesion properties in response to temperature changes, a remarkable 7–8 °C reduction in the critical solution temperature was obtained. Considering the faster print time, the surface finishing and printing resolution of the 3D objects printed with DLP printers have suffered [125]. Furthermore, light-induced printing has made it harder to regulate the porosity of printed structures [126].

### 5.2. Microextrusion-Based Printing

This technique has been employed for various applications in tissue engineering, for example, neural, cardiac, cartilage, skeletal muscle, and liver [127–138]. Microextrusion-based printing utilizes a thermally-regulated dispensing system, a video recorder, a piezoelectric humidifier, and a fiber-optic laser source to constantly dispense biomaterials and biological agents via a nozzle attached to the bioink cartridge.

The microextrusion-based technique generates thick vertical structures, enhances cell density, increases viscosity, and facilitates different polymerization mechanisms. Nevertheless, they are susceptible to nozzle blockage, the accomplishing interlayer interaction is complex, and the nozzle shearing could decrease the viability of cells in high-resolution structures [139]. Billiet et al. fabricated a highly porous cell-loaded GelMA framework by utilizing a microextrusion-based printing method for the application in tissue engineering and achieved an increase in the viability of cells up to 97% due to the printing accuracy of the construct compared to their previous findings [140,141]. Microextrusion printers effectively used nanocellulose-alginate biopolymers to generate human chondrocytes-loaded composites that sustained remarkable proliferation and cell survival under in vitro culture. This suggests that the nanocellulose-based bioink could be employed to construct articular cartilage tissue [142].

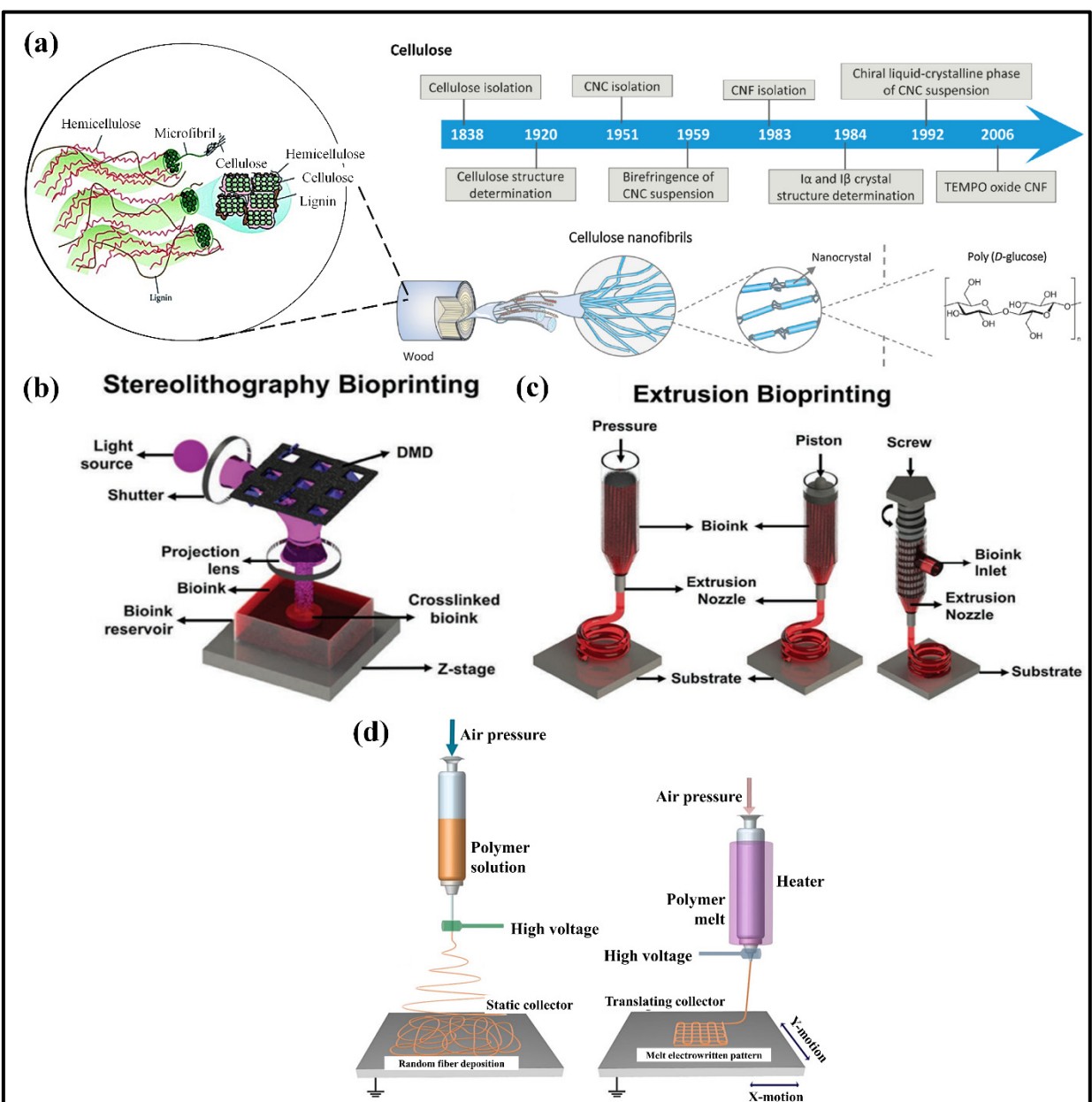

**Figure 3.** (**a**) Cellulose nanocrystals discoveries milestone in the structural characterization and preparations. Adapted with permission from Ref. [143]. (**b**) Stereolithography (SLA) printing, (**c**) Extrusion bioprinting. Adapted with permission from Ref. [144]. (**d**) Solution and melt electrowriting technique. Adapted with permission from Ref. [145].

### 5.3. Electrowriting Based on Melt and Solution

Smooth scaffold composites with a relatively small fiber size, identical to the extracellular matrix (ECM) of natural tissues, can be manufactured via melt and solution-based electrowriting techniques [146,147]. In the direct writing electrospinning process, a solution of polymers is electrospun on a collector or a moving platform with a specified X–Y translation. The fibers that have been collected can be bundled together to form a 3D structure. Chen et al. were the first who described this technology and employed this method to make composites that resembled the articular cartilage's zonal pattern. When human mesenchymal stem cells were cultured on a direct writing, electrospun-fabricated composite, they showed an increase in the chondrogenic marker (Sox9) and Aggercan (ACAN), compared with the cultivation on typical electrospun constructs [146].

Melt electrowriting, abbreviated as MEW, is a technique that integrates classical electrospinning with 3D printing. In this technique, the material is first melted and then spun to a collector under a voltage supply, which dictates the ultimate structure of the manufactured product. Melt electrowriting yields fiber with a larger diameter than the fibers obtained with direct writing electrospinning, typically within 5–15 μm [147]. According to the studies, this technique has produced scaffolds for myocardial tissue, vascular tissue, and heart valves [148–150]. Table 1 summarizes the nanocellulose-based composite's outcome for the healing and regeneration of various tissues.

**Table 1.** The positive results of the nanocellulose-based composites.

| Composites Combined with Cellulose and Natural Fibers | Source of Matrix | Accepted Chemical Methods | Significant Outcomes | Ref. |
|---|---|---|---|---|
| Cellulose nanofibers (CNFs) | Epoxy/Diglyceryl Ether of Bisphenol (DGEBA) | - | The presence of CNFs significantly enhances the nanocomposites' storage, thermal, and loss modulus. | [151] |
| Cellulose nanofibers, Bacterial nanocellulose (CNFs/BNC) | Polyvinyl Alcohol (PVA) | Acetylation | CNFs have enhanced mechanical characteristics and remarkably visible light transmission. | [152] |
| Microcrystalline cellulose (MCC) | Polyvinyl alcohol (PVA) | NaOH/Urea | PVA-based MCC has improved fracture toughness. | [153] |
| Lignin-coated cellulose nanocrystals (L-CNCs) | Polylactic acid (PLA) | - | Thermo-mechanical characteristics are enhanced. | [154] |
| Sugarcane Bagasse and cellulose nanocrystals (SCB/CNCs) | k-carrageenan | Alkali | The incorporation of CNCs increased the mechanical strength. | [155] |
| Cellulose nanofibrils | - | - | The cellulose nanofibrils increased the compression resistance to deformation. | [156] |

### 5.4. Electrospinning

Electrospinning is a broadly adopted nanofiber manufacturing technique to produce cellulose-based composite nanofibers [157]. The technique utilizes a high-voltage supply to produce a liquid jet. In this technique, solid fibers are produced as an electrified jet composed of a high-viscosity polymer mixture and are continually extended via the electrostatic repulsion among solvent evaporation and surface charges [158]. Nanofibers are produced when a liquid polymer mixture is introduced to a strong electric field through a syringe needle or capillary tube. A Taylor cone is generated when the electrostatic forces overcome a liquid's surface tension, and it causes the thin jet to be swiftly accelerated towards various collecting plates. Instability in the jet causes a whipping motion that lengthens and narrows the jet, enabling some solvent to evaporate or cooling the melt to obtain nanofibers on collecting plates. As a result, random non-woven films, electrospun nanofibers, and uniaxially aligned sheets are produced [159]. The nanofibers produced by the electrospinning method have a broad range of applications in the medical field. Incorporating antimicrobial agents, nanoparticles, and drug molecules into nanofibrous dressings can lower the chances of infection and can be utilized as a potential candidate to reduce inflammation and antibacterial properties [160,161].

### 6. Biomedical Applications of Nanocellulose

#### 6.1. Replacement of Blood Vessels

Heart bypass surgery, conducted to deliver blood to the cardiac tissue with an appropriate replacement of blood vessels, is among the most popular therapies for cardiovascular illness. Nanocellulose (particularly BNC) can be developed as a medium for synthetic conduits that can substitute moderate or larger vascular grafts (Figure 4a) because of its biocompatibility and mechanical characteristics. Dieter Klemm's team at the University of

Jena in Germany was the first to investigate and utilize engineered circulatory substitutes made from bacterial nanocellulose nanomaterials [162–164]. They have presented a therapeutic product called Bacterial Synthesized Cellulose, also known as BASYC. The BASYC has shown excellent moisture retention, good mechanical strength in wet conditions, and reduced internal tube surface roughness. BASYC from the bacterial nanocellulose has been effectively employed as a synthetic blood vessel in pigs and rats for microsurgery [165,166].

Unlike the BNC biosynthetic method, manually constructing tubes from CNF and CNC is challenging. As a result, a matrix material is frequently used in developing CNF and CNC-dependent blood vessel transplants. Brown et al. described the manufacture of CNC-based (fibrin/CNC) biocomposites for small-sized vascular graft replacement. CNC was covalently bonded onto a fibrin matrix, providing nano-reinforcement concerning the elasticity and strength of the composite material [167].

Biopolymers made of polyurethane and CNF are reported as promising materials for blood vessel replacement. The inclusion of the CNF in polyurethane increased the material's elasticity and mechanical and physical characteristics. In an adult male patient suffering from multiple endocrine neoplasia type 2B (MEN 2B), polyurethane/CNF biomaterials with 0.7–1.0 mm wall thickness were used as vascular prosthesis between the right carotid artery and the brachiocephalic trunk. Figure 4b shows the vascular prosthesis implants of polyurethane and CNF to treat MEN 2B. Admittedly, no other diagnostic and therapeutic effects of this polyurethane/CNF biomaterial have been recorded [7].

### 6.2. Bone Tissue Engineering

Sukul et al. investigated the long-term secretion of bone morphogenic protein (BMP) and vascular endothelial growth factor (VEGF) through the nanocellulose scaffold for the regeneration of bone tissues [168]. Proliferative and adhesive properties were improved using the nanocellulose-based scaffolds. The bone morphogenic protein-2 (BMP2) and vascular endothelial growth factor (VEGF) loaded scaffolds showed an increase in the orthopedic deformities of rat bone marrow stem cells (RBMSCs), either with or without the treatment of stem cells. Following the 7 and 14 days of the treatment, the growth factor's influence on the alkaline phosphatase protein (ALP) expression was observed (Figure 4c). The BMP2 and VEGF growth factors' loaded constructs showed an increased ALP expression compared to the control, implying that it has a greater bone repair and regeneration ability. For the human bone marrow-derived mesenchymal stem cells, gelatin/alginate scaffoldings incorporated with CNC outperformed pure gelatin/alginate scaffolds in cellular activity. CNC's presence in the polymer matrices significantly influenced cellular functions [169]. Compared to gelatin/alginate scaffold, CNC containing gelatin/alginate scaffold showed improved swelling properties and mechanical strength. For BMSCs, when cultured on CNC/Chitosan and chitosan scaffolds, the CNC/chitosan composites resulted in enhanced osteogenic potential and cellular functions when compared with scaffolds containing only chitosan. Figure 4d depicted the fluorescence microscopy images of BMSCs on the CNC/chitosan composite. Sarkar et al. synthesized hydroxyapatite/carboxymethyl cellulose nanostructures for drug delivery and osteogenesis. To track the medication distribution in cells and tissues, carbon dots were incorporated into the hydroxyapatite/carboxymethyl nanostructures [170].

### 6.3. Biosensors

Nanocellulose is an excellent material for immobilizing biological molecules, which is helpful in diagnostic and biosensor applications. For example, a one-step bio-template approach was used to make a composite of gold BNC in the aqueous solution. The conductivity and biocompatibility displayed by this composite were found to be remarkable. Its fine nanofiber architecture can trap and preserve the function of an enzyme called Horseradish peroxidase (HRP). Biosensors based on the HRP enzyme can detect hydrogen peroxide at concentrations lower than 1 μm [171]. Layers of nanocellulose have been used for antibody immobilization; the layers are carboxylated first and then activated

with proteins. Physical adsorption allows this activated layer to capture antihuman immunoglobulin G antibodies [171,172].

Copolymer grafts can potentially be used to activate the nanocellulose layer. A peptide having a high affinity for human antibodies (IgG) is attached to the grafted polymer to improve the selectivity and specificity of the nanocellulose layer. The immobilization of nanocellulose with growth factors and proteins is used to improve biocompatibility. Sequences of amino acids have been used to strengthen the adhesive properties [173,174].

Nanocellulose, on the other hand, is also used in electrochemical sensors for detecting DNA hybridization. Nanocellulose made from cotton was used to insert DNA. Compared to the traditional nanocellulose, the DNA-embedded nanocellulose agglomerates in significantly greater sizes. Using a reducing agent of sodium borohydride, researchers were able to deposit silver nanomaterials on the nanocellulose. The presence of nanocellulose avoided the agglomeration [175].

### 6.4. Wound Healing

Hemostasis, inflammation, granulation, proliferation, maturation, or remodeling are all processes that occur during wound healing. Consequently, materials based on nanocellulose are potentially used to address each stage by acting as an antimicrobial, anti-inflammatory, or hemostat agent; the incorporation of growth factors and cytokines is allowed to encourage angiogenesis and re-epithelization. For wound dressing and wound healing applications, the basic needs should be met by the desirable nanocellulose materials such as (a) ought to be non-toxic and biologically active, devoid of anaphylactogen and pyrogen; (b) highly permeable for the uptake of blood, diffusion of the drug, and gas transmission; (c) competent for controlling the microenvironment (pH, moisture, and temperature) in the wound area; and (d) effective against the pathogen and infection [176,177].

Because of its exciting and unique characteristics, such as high biocompatibility, excellent water holding capacity, and high purity, bacterial cellulose is intensively researched and utilized as a compatible biomaterial for wound healing and repair. Figure 4e,f show the potential application and wound healing rate using nanocellulose bio-nanocomposites.

Nanocellulose is used to patch acute burns with artificial skin, regulate chronic wound healing with dressing, and mold synthetic blood vessels in reconstructive surgeries. Integrating biologically active substances, including inorganic salts, chitosan, antibiotics, organic metals, or natural plant products, can provide antibacterial properties [178,179]. Other forms of nanocellulose, such as CNC and CNF, were also investigated for potential application in wound dressing and healing, but often in composite or modified forms. For example, a multipurpose hydrogel was constructed using genipin crosslinking diosgenin and gelatin to make a semi-IPN composite. The hydrogel showed non-cytotoxicity, high mechanical characteristics, and effective antimicrobial capabilities, which are essential in the healing process to speed up wound healing and avoid infection [180].

Nanocellulose-based scaffolds with a controllable stiffness were printed by using a 3D printer and crosslinked by employing a dual crosslinking approach, a chemical crosslinking with 1,4-butanediol diglycidyl ether and in situ calcium ($Ca^{2+}$) crosslinking, to govern the rigidity of the scaffold, which has been found to influence the cellular behavior during wound healing [181]. A dual crosslinking approach involving UV crosslinking and in situ $Ca^{2+}$ crosslinking was developed to print skin patches to provide excellent printability while using low nanocellulose inks. The nanocellulose-based hydrogel integrated gelatin methacrylate was reported to aid the design process and encourage the proliferation of cells [182].

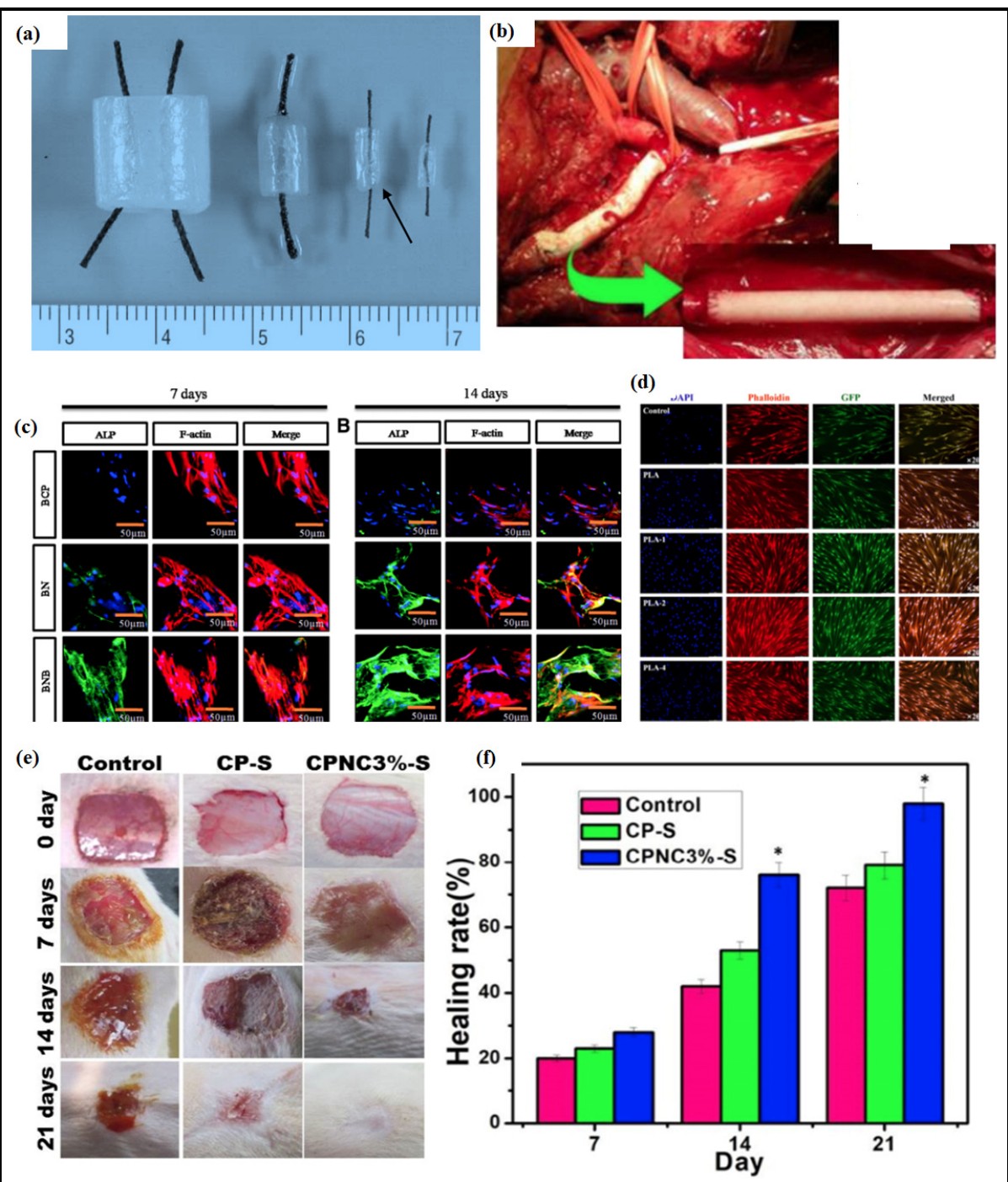

**Figure 4.** Nanocellulose-based biomedical applications. (**a**) Different sized bacterial synthesized cellulose tubes. Adapted with permission from Ref. [163]. (**b**) Vascular prosthesis implant consisting of polyurethane/nanocellulose for treating multiple endocrine neoplasia 2B. Adapted with permission from Ref. [183]. (**c**) Confocal micrograph of localization of ALP protein (green) in RBM-SCs after treatment with nanocellulose-based scaffold for 7 and 14 days to assess the effect of VEGF and BMP2 growth factors. (**d**) Fluorescence microscopy images of bMSCs growing on chitosan/CNC nanofibers. Adapted with permission from Ref. [184]. (**e**) Images of wound treated with control, CP-S (Chitosan/polyvinyl pyrrolidone-stearic acid), and CPNC 3% (Chitosan/polyvinyl pyrrolidone/nanocellulose-stearic acid 3%). (**f**) The wound healing rate of control, CP-S, and CPNC3%-S bio-nanocomposite. * Considerably difference compared to the control group ($p < 0.05$). Adapted with permission from Ref. [185].

## 7. Conclusions and Future Perspective

Nanocellulose has proven to be one of the most popular environment-friendly materials for various uses, attracting interest in areas from scientific research to industrial use. The current review study examines the potential of nanocellulose as a viable biocompatible material. The manufacture and modifications of nanocellulose have been the subject of extensive investigation. Because of its eco-friendly and sustainable nature, mechanical treatment and enzyme hydrolysis are popular methods to obtain nanocellulose. The characteristic features of a variety of nanocellulose such as CNC, CNF, and BNC can be significantly altered by surface functionalization. The surface-modified nanocellulose and nanocellulose-based composite materials have a broad range of biomedical applications, including blood-vessel replacement, bone regeneration, wound healing, and in biosensing applications.

However, there are still several issues that need to be resolved before the use of nanocellulose in therapeutic settings. For instance, the biosafety of nanocellulose and its biomaterials must be thoroughly assessed using appropriate methodologies and clinically meaningful procedures. The in vivo and in vitro evaluation of biodegradation patterns of nanocellulose should be managed appropriately and studied accurately. Future research should focus on the toxic effects of additives commonly used to improve the dressing's effectiveness and may adversely affect the healing process. The mechanical properties of the scaffolds can be significantly improved after incorporating nanocellulose. Despite this, the human system has difficulty degrading nanocellulose materials, and the interaction mechanism between cells and nanocellulose remains unknown. From a future perspective, it is essential to investigate the complications associated with nanocellulose and its biomaterials for its utilization in tissue engineering and several other biomedical applications.

**Author Contributions:** Conceptualization, methodology, and software, A.R.; validation, formal analysis, and investigation, S.D.D.; resources and data curation, K.G., writing—original draft preparation, A.R.; writing—review and editing, A.R. and T.V.P.; visualization, D.K.P.; supervision, K.-T.L.; project administration, S.D.D. and K.-T.L.; funding acquisition, K.-T.L. All authors have read and agreed to the published version of the manuscript.

**Funding:** This research was supported by the Basic Research Program through the National Research Foundation of Korea (NRF), funded by the Ministry of Education (No. 2018R1A6A1A03025582, No. 2019R1D1A3A03103828, and No. 2022R1I1A3063302), Republic of Korea.

**Institutional Review Board Statement:** Not applicable.

**Informed Consent Statement:** Not applicable.

**Data Availability Statement:** Not applicable.

**Conflicts of Interest:** The authors declare no conflict of interest.

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
