# Peer review of "A Review of Properties of Nanocellulose, Its Synthesis, and Potential in Biomedical Applications"

_applsci, doi:10.3390/app12147090_

Round 1

Reviewer 1 Report

I am happy to review the review on "A review of properties of nanocellulose, its synthesis, and potential in biomedical applications" The article is neatly written with appropriate justifications. However, I wonder why authors have not given any scope to discuss on the blends/composites of the nanocellulose? It is suggested to add the literature pertaining to nanocellulose blends together with their applications. 

Electrospinning is another important technique where many researchers have put their efforts to obtain the fibres. By adding fibres obtained from such techniques may instigate and attract more researchers to the article.

Reviewer 2 Report

Please see the attached

Round 2

Reviewer 1 Report

Accept